# TunR2, a novel mode-of-action tunicamycin-type antibiotic: Pharmacokinetics in C57BL/6 mouse and Holstein cattle

Maria A. Colombatti Olivieri[1,2,3]*, Eric D. Cassmann[1], Michael A. Jackson[4], Neil P. J. Price[4], John P. Bannantine[1]

1 USDA-Agricultural Research Service-National Animal Disease Center, Ames, Iowa, United States of America, 2 Oak Ridge Institute for Science and Education (ORISE), ARS Participation Program, Oak Ridge, Tennessee, United States of America, 3 Instituto de Agrobiotecnología y Biología Molecular (IABIMO), INTA-CONICET, Hurlingham, Buenos Aires, Argentina, 4 USDA-Agricultural Research Service-National Center for Agricultural Utilization Research, Peoria, Illinois, United States of America

* macolombatti@gmail.com

## Abstract

We have investigated the pharmacokinetics of TunR2, a modified tunicamycin-type antibiotic, in mice and cattle. TunR2 has previously been shown to be effective in a mycobacterial disease model using zebrafish, with a minimal activation of the eukaryotic unfolded protein response (*upr*) and a reduction in the *in vivo* mycobacterial burden. In this study, we presented statistically relevant pharmacokinetics of native tunicamycin (Tun) and two less toxic modified analogs, TunR2 and TunR1, using a well-defined clonal C57BL/6 mouse (both male and female). Blood samples were collected at multiple time points, and plasma concentrations were quantified using liquid chromatography-tandem mass spectrometry (LC-MS/MS). Pharmacokinetic parameters were calculated using a two-compartment analysis. Our findings indicate that Tun and TunR1 tend to distribute in tissue compared to TunR2, which has a longer half-life than Tun. This translates into longer TunR2 activity time, potentially allowing for less frequent dosing than Tun or TunR1. We subsequently administered the modified TunR2 to Holstein cattle using a three-bolus intravenous regimen. We monitored blood, milk, urine, and feces over 90 days. In dairy cattle, the pharmacokinetics of TunR2 appear to be cumulative, and clear after 10 days. These findings provide critical new insights into the pharmacokinetics of TunR2. We concluded that TunR2 has considerable potential for treating bacterial infections, particularly as an antimicrobial adjuvant with well-established β-lactam antibiotics. Further studies are required to study safety and optimize dosing regimens for effective therapeutic use, as well as in combination with other antibiotics, such as β-lactams.

purpose. The work is made available under the Creative Commons CC0 public domain dedication.

**Data availability statement:** All relevant data are within the manuscript and its Supporting Information files.

**Funding:** The study was supported by the USDA Agricultural Research Service (internal research project #12334 to John P. Bannantine). The funders had no role in study design, data collection and analysis, decision to publish, or preparation of the manuscript.

**Competing interests:** The authors have declared that no competing interests exist.

## Introduction

Antimicrobial resistance (AMR) is a growing global public health crisis that threatens the effectiveness of antibiotics, posing significant risks to humans and animals [1]. AMR occurs when bacteria become resistant to drugs, making common infections harder and more expensive to treat. In veterinary medicine, resistant pathogens complicate the treatment of infectious diseases in livestock and companion animals, potentially affecting productivity and food security, as well as enabling the transmission of resistant bacteria to humans [2]. The discovery of new antimicrobial agents is essential, especially given the slow pace of new antibiotic development in recent decades [3].

Natural tunicamycin (Tun) is produced by fermentation in several *Streptomyces* species and is used commercially on a large scale. It inhibits the polyprenyl-P *N*-acetylhexosamine-1-P transferase (PNPT) enzyme, which is part of a superfamily. In eukaryotes, the PNPT homolog DPAGT1 catalyzes the first step in protein *N*-glycosylation, and Tun inhibition results in the misfolding of glycoproteins and a lethal unfolded protein shock cellular response. Toxicity of this drug has been reported in different animal models, including rodents, cattle, sheep, and pigs, where treatment with Tun produces mainly neurological signs, hepatotoxicity, and kidney lesions [4–7].

TunR1 and TunR2 are a new type of nucleoside antibiotic developed by the USDA based on the natural product tunicamycin (Tun). TunR1 and TunR2 have a reduced trans-2,3 double bond in the N-acyl chain, and in TunR2, the tunicaminyl N-uracil group is also replaced by N-linked 5,6-dideoxyuracil (Fig 1). These compounds are

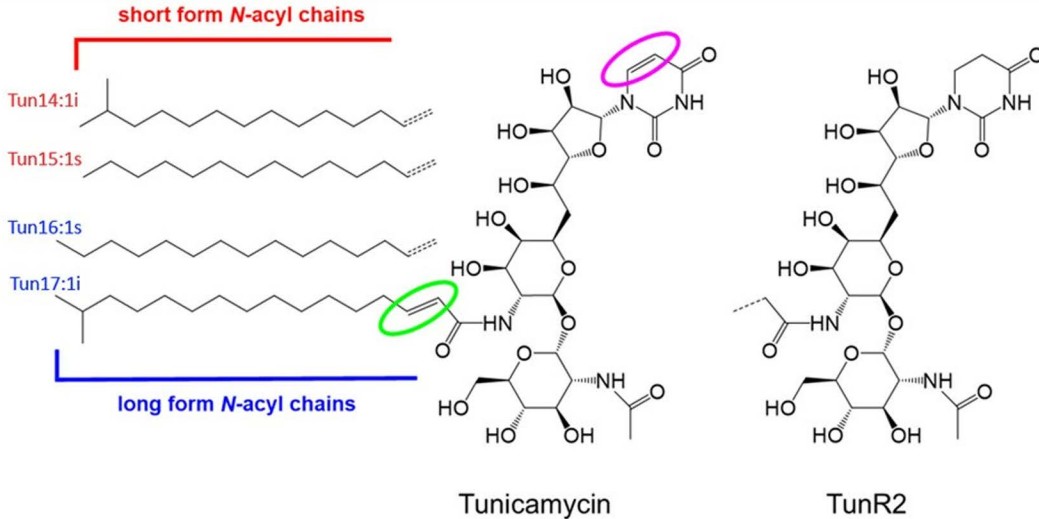

**Fig 1. Comparative structures of TunR2 (right) and native tunicamycin (center) showing the N-acetylglucosamine-tunicamine-uracil head group.** The various N-linked acyl groups are shown on the left. The modified TunR2 differs from native Tun in that both double bonds (designated by colored circles) are reduced, with the planar uracil ring being converted to a non-planar 5,6-dihydrouracil (DHU). The non-planarity of the DHU group in TunR2 results in attenuated π-π binding to an active-site Phe residue in the eukaryotic PNPT/GPT target protein, which is the mechanism for the reduced mammalian toxicity of the TunR2 molecule.

considerably less toxic to eukaryotes whilst maintaining potent adjuvant and antimycobacterial properties [8,9]. The TunR2 analogs have significantly lower eukaryotic toxicity compared to native Tun in *Saccharomyces* yeast, eukaryotic cell lines (Chinese hamster ovary CHO and human MDA-MB-231) [8], a live insect model (fall armyworm, *Spodoptera frugiperda*) [9], and embryonic zebrafish (*Danio rerio*) [10]. TunR2 has also shown efficacy in a mycobacteria/zebrafish disease model, with attenuation of the host unfolded protein response (*upr*) and lowering of the mycobacterial infection and the concomitant macrophage response [10]. Although these two synthetic compounds have shown reduced toxicity compared to the *Streptomyces*-produced Tun, this observation needs to be further assessed in mammals. Finally, they are also potent adjuvants for β-lactams, some cases enhancing their activity by more than100-fold [11,12].

Natural Tun is not a single compound, but rather, it exists as a mixture of various *N*-acylated forms that typically have either a C14, C15, C16, or C17 carbon chain length fatty acid group attached to the core tunicamine head group (Fig 1). These *N*-acyl groups generally contain an unusual *trans*-2,3-double bond and also have a characteristic branching pattern at the end of the acyl chain (either *iso*, *anteiso*, or *straight* chain), giving rise to multiple forms of the Tuns in commercial preparations, and hence also in the modified TunR1 and TunR2 made from these stocks.

In the present work, we evaluated the pharmacokinetics (PK) of Tun, TunR1, and TunR2 administered intravenously to small and large mammals (mice and cattle) and demonstrated that their clearance occurs primarily through the liver and feces via biliary excretion. When TunR2 is solubilized with deoxycholate (DOC) in aqueous-based solution, it can be administered via three bolus injections in cattle. We report the duration and route of TunR2 through mice and dairy cattle. We further demonstrated that the TunR2 is metabolically unchanged and is excreted from these animals without apparent chemical modifications.

## Materials and methods

### Chemicals

The Tun metabolites used were isolated from liquid cultures of *Streptomyces chartreusis* NRRL B-12338 obtained from the USDA Agricultural Research Service Culture Collection, Peoria, IL. TunR1 and TunR2 were prepared from the native Tun by selective catalytic hydrogenations, as described previously [13–15]. All other chemicals and solvents were purchased from Sigma-Aldrich, St. Louis, MO. The Tun, TunR1, and TunR2 were partially purified by fractionation on reversed-phase SPE cartridges (Waters Inc.) using a 60–100% aqueous methanol elution gradient. Tun-containing fractions were identified in the mass range 800–1000 Da by MALDI/MS and stored at −20 °C. Further purification was attained by reverse-phase HPLC fractionation as described below.

### C-30 Reverse Phase HPLC and MALDI-TOF MS analysis

The HPLC system used was a Finnigan Surveyor (ThermoFisher Scientific, W. Palm Beach, FL) using a reversed-phase C30 column specifically designed to separate long-chain carotenes (YMC Carotene C-30, 4.6 x 250 mm; 3 μm particle size) (YMC America Inc., Allentown, PA). The Tun *N*-acyl variants were resolved using an acetonitrile-water gradient (45–100%; 1 mL/min; 27 °C) and typically eluted from the column in order of increasing acyl chain length between 5–11 min. The peaks were monitored by PDA detector and were collected manually from multiple injection runs. The Tun components were verified by molecular mass using MALDI-TOF/MS analysis on a Bruker-Daltonics Microflex instrument (Bruker-Daltonics, Billerica, MA, SA) as described previously [13]. Briefly, samples were diluted 1:1 v/v with 2,5-dihydroxybenzoate matrix (saturated solution in acetonitrile; typically, 1−2 μL) and spotted onto a standard 100-place stainless steel target. The MS spectra were acquired in reflectron mode with positive ion detection, averaging 2000 shots at 60–70% laser power (150 μJ maximum output, 337 nm). A pulsed ion extraction (200 ns) was used with the ion sources set to 19.0 kV and 13.6 kV, respectively. The purity of the resulting TUN metabolites was verified by LC-ESI/MS using an Agilent 1290 Infinity II System with an Agilent MSD mass spectrometer with in-line flow from the RP C30 HPLC column described above. The solvent gradient used consisted of solvent A (0.1% formic acid in HPLC-grade water) and solvent B (0.1% formic acid in

acetonitrile). Metabolites were detected in positive ion electrospray mode by the MSD using nitrogen as the flow gas (350 27 °C; 13 L.min$^{-1}$; nebulizer 60 psig). Full MS spectra were acquired for the mass range $m/z$ 150–2000 for all samples. Where appropriate a voltage ramp was used to induce ion fragmentations as described previously [11].

## Standard curve

For Tun, TunR1, and TunR2 quantification, a partial validation was performed using mice serum and cow biological samples (including blood, milk, urine, and feces) as the matrix. A standard curve in water was also used, including a negative control. The standard stock solution of Tun, TunR1, and TunR2 was prepared at a concentration of 10 mg/mL in DMSO for the mouse trial. Additionally, TunR2 was prepared at 10 mg/mL plus 5 mg/mL of DOC in water for the cow trial. These solutions were stored at −20°C.

Serial dilutions of the stock solution were made to prepare drug standard curves in water and spiked biological samples in a range from 0 to 150 ug/mL. In addition, quality control samples (QC) were evaluated in each type of biological sample. These QC samples consisted of drug-free samples fortified with Tun, TunR1, or TunR2 at low, medium, and high concentrations with a high linearity between peak areas and concentrations on the Photodiode Array Detector (PDA) response ($R^2 = 0.9994$). The limit of detection (LOD) was taken as signal-to-noise of 3:1, being 10 ng/mL in water samples and 20 ng/mL in biological samples.

Samples were dried to concentrate them, and the dried residues were resuspended in methanol, which was filtered through a cotton layer into a GC insert inside a capped LC vial. The MeOH filtrate was injected onto a C30 RP column using an aqueous AcCN gradient and then analyzed by LC-SIM/MS. Concentration curves for each $N$-acyl variant were calculated based on the peak area of the analyte versus concentration. This was done using GraphPad by performing least squares linear regression, with the line forced through zero ($R^2 > 0.9540$).

## Protein binding

The binding of Tun, TunR1, and TunR2 to bovine serum albumin (BSA) was determined *in vitro* using equilibrium dialysis. The compounds were prepared from a 10 mg/mL stock solution in DMSO or DOC, as previously described. Plasma protein binding studies were conducted in triplicate using 96-well Rapid Equilibrium Dialysis (RED) inserts and plates (Thermo Scientific™).

For each well, an aliquot (100 µl) of drug-spiked BSA solution (at 35 mg/mL in PBS) was added to the sample side of the chamber. A 350 µl volume of PBS buffer (containing 0.002% Tween 80) was added to each well's buffer chamber side. The plate was covered with adhesive sealing film to prevent evaporation and placed at 37°C for 4 h with a shaking speed of 200 rpm. After incubation, the samples were processed and analyzed by LC-SIM/MS, as described above.

## Animals

The mouse PK study was conducted by the preclinical research services at The Jackson Laboratory (JAX), Bar Harbor, ME, with the LC/MS and MALDI-TOF/MS analysis performed at the USDA National Center for Agricultural Utilization Research (NCAUR), Peoria, IL. Tunicamycin (Tun), TunR1, and TunR2 were prepared at NCAUR and shipped as dry solids (4 mg of each) in small glass vials. Dimethylsulfoxide carrier (DMSO, 400 uL) was added to the tubes to give 10 mg/mL stock solutions and stored at 4 ºC as required. Six to nine-weeks-old male and female C57BL/6 (JAX stock# 000664) mice were used for the PK study. The mice were ear-notched for identification and housed in individually ventilated polysulfone cages with HEPA-filtered air at a density of up to 5 mice per cage. Cages were changed every two weeks. The animal room was lighted with artificial fluorescent lighting in a controlled 12 h light/dark cycle room (6 am to 6 pm light). The temperature and relative humidity ranges in the animal rooms were maintained at 20–26°C and 30–70%, respectively, and were set to have up to 15 air exchanges per hour. Filtered tap water, acidified to a pH of 2.5 to 3.0, and standard lab chow

was provided *ad libitum*. The animals were euthanized by CO2 asphyxiation, followed by cervical dislocation, at the end of the study (24 h).

For the cattle pilot trial, four Holstein cows were used in this study. These cows, with ages ranging from 6 to 10 years, were housed outdoors on pasture or in the on-site dairy barn for the duration of the study. Animals 5327, 6739 and 6878 were treated with TunR2 while 12802 was treated only with the solubilizing carrier deoxycholic acid (5%). The use of deoxycholate (DOC) circumvented the need for DMSO as the carrier, making it safer for use in cows. Tissues from other control cows (not treated with TunR2 or DOC), 5466, 2222, 1307, and 1422, were obtained from our frozen repository of stored samples and used for comparative purposes. All cows, except #1422, had Johne's disease (JD) as indicated by the test results including IDEXX ELISA antibody test, IFN-γ (bovigam) test and IS900 fecal PCR, and in some cases with observable disease signs (cow #6878). Also, all cows were positive for bovine leukemia virus (BLV), which is highly endemic. The standard feed included a mixed pelleted ration and hay cubes which were provided throughout the study. Body condition scores were measured for several months prior to the start of the study and at the study conclusion. At the end of the study, the animals were euthanized via intravenous administration of sodium pentobarbital.

### Ethics statement

This study followed the principles of the 3Rs, focusing on reducing pain and distress. All mice experiments were performed according to the protocols approved by the Institutional Animal Care and Use Committee of the JAX and the NIH Guide for the Care and Use of Laboratory Animals. The cattle pilot trial was approved by the NADC Institutional Animal Care and Use Committee (protocol # ARS-22–1050, NADC), most recently approved in January 2023, in line with the guidelines set by the Animal Welfare Act. Housing facilities are accredited by the American Association for Accreditation of Laboratory Animal Care. The animals were provided with environmental enrichment to enhance their well-being, and humane endpoints were established to ensure prompt euthanasia if their condition deteriorated beyond an acceptable point.

### The pharmacokinetic profiles of tunicamycin, TunR1, and TunR2 in C57BL/6 mice (1st trial)

Eighty, 40 male (M) and 40 female (F), C57BL/6 mice were enrolled in the study on Day 0. Body weights, clinical observations, and digital caliper measurements were recorded. Eight (4M/4F) animals per treatment received a single 30uL intravenous bolus dose of the experimental samples (or control DMSO carrier) via the lateral tail vein, as shown in Fig 2A and S1 Table. Essentially, the control group 1 received DMSO, groups 2–4 received the tunicamycin stock solution in DMSO (0.2, 2.0, and 10 mg/mL), groups 5–7 received TunR1 stock solution in DMSO (0.2, 2.0, and 10 mg/mL), and groups 8–10 received TunR2 stock solution in DMSO (0.2, 2.0, and 10 mg/mL). After the first dosing, blood draws for the PK study were at 8-time points (5 min, 15 min, 30 min, 1 h, 2 h, 4 h, 6 h, 24 h). The mice were subdivided into 2 groups of 4 (2M/2F) mice each, with each subgroup having ~50 ul of blood collected via retro-orbital bleed for 4-time points (subgroup 1: 5 min, 30 min, 2h, and 6h; subgroup 2: 15 min, 1h, 4h, and 24 h). In life, a collection of 50 μL of blood was diluted with 450 μL of ethanol and evaporated to dryness on a Speedvac evaporator under sterile conditions. The dried residues were stored at −80 ℃ and were shipped on dry ice to NCAUR, Peoria, for analysis. At the end of the study (24 h), the animals were euthanized, and terminal blood samples from all mice (100 μL) were collected via cardiac puncture, diluted with 900 μL of ethanol, and evaporated to dryness on a Speedvac. The dried residues are stored at −80 ℃ until ready for analysis.

For the comparative PK of tunicamycins in blood, drug concentrations were measured by LC–MS/MS as previously described. Different compartmental models (0, 1, or 2 compartments) and weighting options were assessed for goodness of fit by visual inspection of the observed versus predicted plots, weighted residual versus observed and predicted plots, and comparison of Akaike Information Criterion (AICc) values. A two-compartment IV bolus model was determined to describe the data best (S1 Fig).

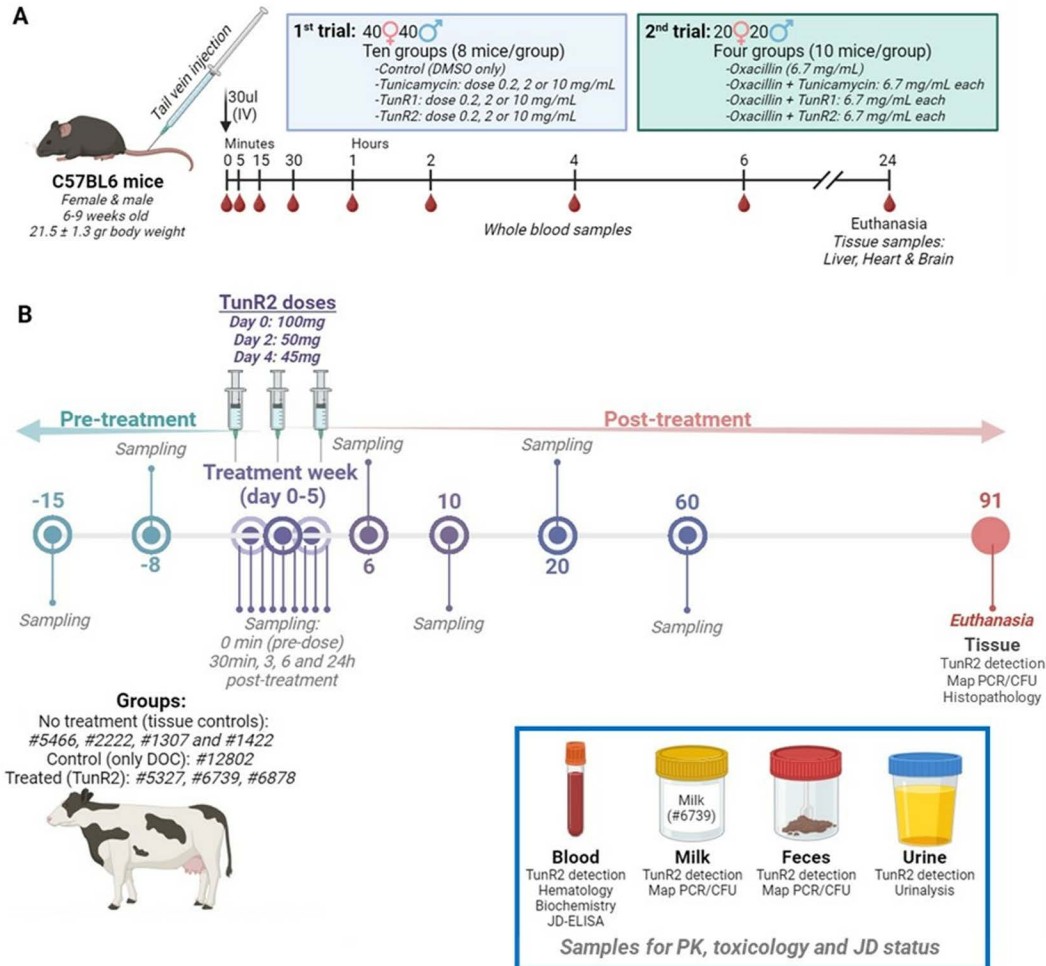

**Fig 2. Animal trial schematic summaries A) Pharmacokinetics of Tun, TunR1, and TunR2 in C57BL/6 clonal mouse.** Two trials were conducted in mice. In the first trial, animals received different doses of Tun, TunR1, or TunR2. In the 2nd trial, animals received the co-administration of the relevant tunicamycin (Tun, TunR1 or TunR2) plus a beta-lactam drug (oxacillin) (see S1 and S2 Tables). Each animal received a single dose (30 µL) of the drug administered intravenously via lateral tail vein injection. Whole blood samples (50 µL) were collected 5 min, 15 min, 30 min, 1h, 2h, 4h, 6h, and 24h post injection. Liver, heart, and brain samples were collected at the final-time point (24 h). B) Experimental timeline summary for TunR2 intravenous pharmacokinetics in Johne's-infected Holstein cattle. Three bolus injections of 100, 50, 45 mg aqueous TunR2/DOC intravenous (jugular vein) were administered to four cows over a period of 5 days. Six samples were collected daily of blood, urine, feces, and milk (Cow 6739 was lactating) for each cow for a period of 90 days and the TunR2 pharmacokinetics (PK) was analyzed by LC-ES/MS and MALDI/MS. At the same time, the health status of the animals was monitored (Johne's disease and possible side effects of TunR2).

Pharmacokinetic parameters were calculated by two-compartmental using a linear regression analysis and residuals method to calculate using Microsoft Excel 2024 using standard equations (Atkinson; 2007; Miyamato et al., 2019). Details about the PK equations to calculate drug concentrations and PK parameters are listed in the S1 Appendix.

The area under the curve from time zero to 24h ($AUC_{0-24}$) were calculated from the predicted concentrations by versus time data using the trapezoidal rule in a bi-exponential function (two phase decay) graph on GraphPad Prism 10 (version 10.3.1; GraphPad Software Inc., San Diego, CA, USA).

For the PK analysis of the different compounds (C15, C16 and C17), the data for AUC and MRT were normalized to a dose of 1 mg/mL of each compound. For this calculation, the ratio of the different compounds that make up Tun, TunR1, and TunR2 was considered.

### Temporal and organ specific co-localization of oxacillin with Tun, TunR1, and TunR2 in C57BL/6 mice (2nd trial)

In the second trial, which was a drug co-localization study, forty (20 male and 20 female) C57BL/6 mice were enrolled on Study Day 0 and were dosed according to the experimental design shown in Fig 2A and S3 Table. Body weights and clinical observations were recorded as before. Essentially, group 1 animals (10, 5 M/5F) received 30uL of oxacillin (6.7 mg/mL), group 2 animals (10, 5M/5F) received oxacillin (6.7 mg/mL) plus tunicamycin (6.7 mg/mL), group 3 animals (10, 5M/5F) received oxacillin (6.7 mg/mL) plus TunR1 (6.7 mg/mL), and group 4 animals (10, 5M/5F) received oxacillin (6.7 mg/mL) plus TunR2 (6.7 mg/mL). Blood samples were taken (50 µL) 5 min, 15 min, 30 min, 1h, 2h, 4h, 6h, and 24h post-injection. At the end of the study (24 h post dose) the mice were euthanized by $CO_2$ asphyxiation, and liver, brain and heart were collected. The organs were homogenized in ethanol, then filtered to remove solids, and the ethanolic filtrate dried by Speedvac evaporation. The dried residues were stored at −80 ºC and shipped on dry ice. Drug concentrations in tissue were measured by LC–MS/MS as previously described.

### Pharmacokinetics of TunR2 in Holstein dairy cattle (a phase 0 pilot study)

Sampling along with weight and temperature of the four cows was taken at −15, −8, and 0 days prior to the administration of the first TunR2 dose and at 0.5 (30 min), 3, 6, and 24 h after each TunR2 dose. Each dose was given intravenously, and animals were observed for 6h to check the absence of anaphylactic or adverse drug reactions. Sampling and measurements continued on days 6, 10, 20, 30, 60, and 91 after the first dose. TunR2 was dissolved in deoxycholate (DOC) at a concentration of 5 mg/mL (Fig 2B). Large animals have lower metabolic rates and require smaller drug doses on a per-weight basis than mice. To compare the 1st dose used in cattle with the higher dose used in mice, a conversion was made with the following formula (Mahmood, 2007; Nait & Jacob, 2016):

$$Animal\ Km = \frac{BW\ (kg)}{BSA\ (m^2)} \tag{1}$$

Where animal Km is the correction factor, BW is body weight, and BSA is body surface area.

$$CED\ \left(\frac{mg}{kg}\right) = Mice\ dose\ \frac{Cattle\ Km}{Mice\ Km} \tag{2}$$

Where CED refers to the cattle equivalent dose.

The cattle equivalent dose, to equal the highest TunR2 dose administered to mice, was 601.3 ± 35 mg/kg. Since we did not evaluate the safety of the drug in the murine model, it was decided to conduct an exploratory Phase 0 clinical trial using microdoses (subtherapeutic doses) to evaluate the PK and potential toxic effects [16]. Therefore, three of the four cows were given the following doses of TunR2: 100 mg/cow on day 0, 50 mg/cow on day 2, and 45 mg/cow on day 4, and one animal was a control that received three doses of the drug vehicle (DOC without TunR2) (S4 Table). These doses correspond to 125 ± 14, 63 ± 7, and 56 ± 6 ug/kg respectively. Therefore, the 1st dose in cattle was 42 ± 2.4 times less than the highest TunR2 dose administered in mice.

Blood, feces, and urine samples were collected at all time points for TunR2 quantification and clinical analysis. Before TunR2 administration, samples were collected to assess the Map burden and evaluate the baseline health status. Feces were cultured in 1-gram aliquots on slants of Herrold's Egg Yolk Medium (HEYM; BBLTM Herrold's Egg Yolk Agar Slants with mycobactin J, amphotericin, nalidixic acid, and vancomycin; Becton Dickinson and Co., Sparks, MD) for Map culture. Quantitative IS900 fecal PCR was also performed. In addition, milk samples from cow #6739 (the only lactating cow) were taken at all time points to determine the presence of Map by IS900 qPCR. A complete blood count (CBC) was performed using Sysmex XN-V Multispecies Hematology Analyzer, using the reference values from Herman et al., 2018 [17], except

for platelet counting [18]. Neutrophil-to-lymphocyte ratio (N:L ratio) and platelets-to-lymphocyte ratio (P:L ratio) were calculated as described by Braun et al., 2021 and Guan et al., 2020 as inflammatory markers [19,20]. Measurements of blood biochemistry were obtained with the Idexx Catalys One Veterinary Blood Chemistry Analyzer using the reference value from the manufacturer, except for albumin-globulin ratio [21] and alanine aminotransferase (ALT) [22]. For urinalysis, density, pH and microscopic examination was evaluated, in addition to urine protein/creatinine ratio (UPC) quantification that was performed at the Iowa State University Clinical Pathology Laboratory and compared to the reference values from Herman et al., 2019 [23].

Animals were necropsied at the end of the 3.5-month study (Day 91), and tissues were collected and either processed immediately or frozen at −80°C (Fig 2B). Tissues included the liver, kidney, spleen, cardiac muscle, lung, mammary gland, uterus, ovaries, intestinal tissues, and associated lymph nodes. The collected tissues were used for histopathology, Map culture and qPCR to assess infection status, as previously described by Jenvey et al. 2018 [24]. For histopathology, all slides were stained at room temperature. Hematoxylin and eosin-stained (H&E) and Ziehl-Neelsen stained tissue sections were examined, including jejunum (proximal, mid, distal), ileum (proximal, mid, distal), intestinal lymph nodes, liver, and spleen. Only H&E staining was performed on the lung, kidney, heart, and mammary gland. For comparison of hepatopathology in cows with paratuberculosis, livers from four additional cows with paratuberculosis that had not received any treatment injections were included in the examination. Histopathologic analysis was performed by a veterinary pathologist (Dipl. ACVP) using an Olympus BX43 light microscope (Evident Scientific, Waltham, MA). Images were captured using a DP28 camera and cell Sens software (Evident Scientific, Waltham, MA).

### Statistical analysis

The normality of the data and the homogeneity of variances were assessed using the Shapiro-Wilk test and Bartlett's test, respectively. For statistical analysis, One-way ANOVA and Tukey's multiple comparisons test was used for comparison of Tun, TunR1 and TunR2 PK parameters, and Brown-Forsythe and Welch ANOVA tests, with Dunnett's T3 multiple comparisons for *N*-acyl variants. A p-value of less than 0.05 was considered statistically significant. Data and graphical representations were generated using GraphPad Prism version 10.0 (GraphPad Software Inc., San Diego, CA, USA).

## Results

### Analytical procedures for Tun, TunR1 and TunR2

Natural Tun, TunR1, and TunR2 are a mixture of various N-acylated forms (Fig 1). It was therefore crucial to have reproducible, quantitative methods with which to determine these various *N*-acyl forms in biological samples. We and others have previously described methods to analyze Tun components based on C18 reversed phase high performance liquid chromatography (RP-HPLC), with detection either by diode array detector or by electrospray mass spectrometry (ESI-MS) [11,12,25]. Electrospray MS of Tun in positive ion mode gives rise to molecular adduct ions (typically $[M+H]^+$ and $[M+Na]^+$) and characteristic fragment ions, the most dominant of which arise from neutral loss of the glucosaminyl moiety to generate $[M+H-221]^+$ ions. We found selective ion monitoring (SIM) of these fragment ions to be the most sensitive method. for monitoring HPLC separations and permitted the detection of Tun components at concentrations as low as 10 ng with a 20 μL HPLC injection loop (Fig 2). Hence, 10 ng of TunR2 in a dry residue from a 240 μL sample of blood is equivalent to a detection limit of <40 μg/L for each of the Tun components (Fig 3).

Matrix-assisted laser desorption/ionization time-of-flight mass spectrometry (MALDI-TOF MS) has previously been used to monitor Tun in biological samples [14,15]. This has the advantage of detecting multiple components simultaneously without the need for prior chromatographic separation. We found that the signal-to-noise background for MALDI-MS was fairly high relative to LC/ES-MS, with the detection limit between 0.1–1.0 ug. However, due to the speed of analysis of the MALDI/TOF-MS we chose to use both techniques (C30 RP-LC-ES-MS and MALDI/TOF-MS) to analyze the biological samples from the mice and cattle studies described below.

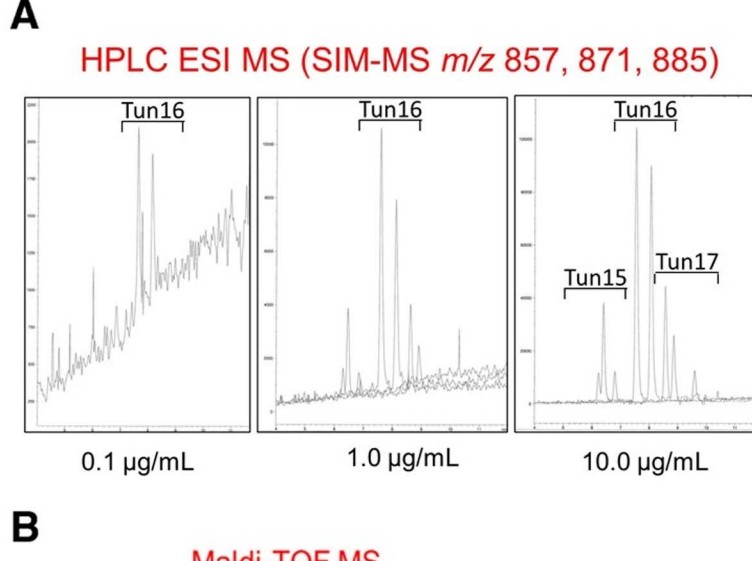

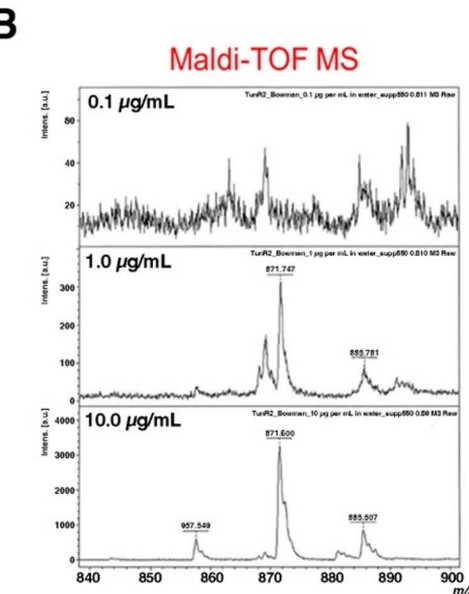

**Fig 3. Comparative detection limits for the most prevalent TunR2 components.** TunR2 exists as a mixture that have C15, C16, or C17 carbon chain length fatty acid groups attached to the core tunicamine head group (Tun15, Tun16, Tun17). This compound was analyzed at 0.1, 1.0, and 10 µg/mL concentrations to determinate the most abundant chain length. A) The HPLC-ESI/MS selective ion monitoring of fragment ions m/z 857, 871 and 885 identifies the N-acyl variants. B) Matrix-assisted laser desorption/ionization time-of flight mass spectrometry (MALDI-TOF MS) also resolves these variants.

### Decreased sensitivity of LC-SIM/MS for Tun detection in blood samples

To enable the quantification of the various Tun components, we initially prepared working stock solutions of Tun, TunR1, and TunR2 in deionized water for comparison with biological samples. The aqueous solubility of these components is low, but we had previously shown the LC detector response to be linear at the low concentration required [15]. However, it immediately became clear that the LC/MS detector response was nonlinear, depending on the nature of the biological samples (S2 Fig). Dried biological samples extracted from cows (either feces, blood, water or milk) were spiked with 1 µg/mL of TunR2 and resuspended in methanol (5 mL) to recover the methanol-soluble TunR2 components. Samples

from these (4 mL) were filtered and dried down on an airline. The blood, milk, and water residues were redissolved in 200 µL methanol, and feces in 1 mL methanol (5x less concentrated compared to the other samples). Samples (20 µL; hence, 80 ng of the original TunR2 spike) were injected into the LC/MS instrument. Four components were observed at 7.5–9.5 min., corresponding to TunR2-16-*iso*, TunR2-16-*straight*, TunR2-17-*anteiso*, and TunR2-17-*iso* (for nomenclature, see Fig 1). It was apparent that these TunR2 spiked components in blood gave peaks that were approximately 5 times less sensitive than those prepared in water, milk, or fecal extracts (S2 Fig). The detection limit in blood was < 40 µg/L for each Tun components (Fig 3). Medium- and long-chain lipids are known to bind to serum albumin, and indeed this is generally recognized as the mechanism responsible for the transport of fatty acids in the bloodstream [26].

Therefore, it was necessary to evaluate the effect of exogenous bovine serum albumin (BSA) on the LC/MS detection of TunR2 in water (S3 Fig). The drug binding to BSA, evaluated by equilibrium dialysis, was estimated at 99.62 ± 0.25% for TunR2 and 99.63 ± 0.06% for Tun (result expressed in Mean ± SD). TunR1 was not detected in the buffer after dialysis, which prevented an estimation of its binding percentage to BSA. It is possible that the concentration of this drug is below the limit of detection (LOD), suggesting that the percentage of binding to proteins may be higher than that of Tun and TunR2 (≥99.8%). A concentration-dependent suppression of the detector response to the TunR2 components was observed, suggesting that the Tun components are indeed attenuated in blood solutions because of binding to serum albumin. Consequently, all subsequent quantitative measurements used defined Tun standards diluted by spiking with the appropriate biological sample under analysis.

## Pharmacology of Tun, TunR1 and TunR2 in C57BL/6 mice

**First trial: Pharmacokinetics of Tun, TunR1 and TunR2.** The overall aim was to determine the pharmacokinetics of modified tunicamycins (TunR1 and TunR2) in comparison to the natural Tun in a well-defined mammalian model (1st mouse trial) and with the co-administration of a beta-lactam drug (oxacillin) in the 2nd mouse trial.

In the first trial, eighty mice were enrolled, comprising 40 females (F) and 40 males (M). Each group included eight mice (4 M, 4F), as shown in Fig 2A and detailed in S1 Table. The mice, aged between 6–9 weeks and averaging 21.5 ± 1.3 g of body weight (b.w.), were identified by ear tags and randomly assigned to the different treatment groups. Each animal received a single intravenous dose of 30 µL at concentrations of 0.2, 2.0, or 10 mg/mL, which corresponded to doses of 0.31, 3.1, or 15.5 mg/Kg of body weight.

At the highest concentration of native tunicamycin (10 mg/mL), one male mouse (from the group of eight) was terminated, while all other treatments were viable. Whole blood samples were taken at different time points and analyzed (by MALDI/MS and LC-ES/MS. Only Tun, TunR1, and TunR2 compounds could be quantified in the blood of the animals that received the highest dose of the drug to calculate the pharmacokinetic parameters.The Tun *N*-acyl components (15, 16, and 17 carbon chain lengths) were detectable by MALDI/MS in the first blood samples drawn (after 5 min), and the *iso*-branched (16i) and straight chain (16s) components, Tun-16, TunR1-16, and TunR2-16, gave LC/ES-MS signals at 7.5–8.5 min retention time suitably resolved for quantification (Fig 4A and 4B). These components were the most abundant, as they were present in higher quantities in the doses administered to the animals. The relative amounts of four different *N*-acylated forms (15s, 16i, 16s, and 17i) of Tun, TunR1, and TunR2 were unchanged in the blood samples over the next 24 hours (S4-A Fig).

The pharmacokinetic parameters for all *N*-acyl variants were analyzed both collectively (Table 1 and Fig 5A) and separately for the C15, C16, and C17 compounds calculated by the bi-exponential curve parameters (Eq. S1) and using Eqs. S2-14 (See S1 Appendix, S5, S6 and S7 Tables). Tun and TunR1 exhibit a high to moderate steady-state distribution volume ($Vd_{ss}$), which was lower for TunR2 (Vdss: Tun > TunR1 > TunR2). Also, the estimated blood drug concentration at time zero ($Cp_0$) is significantly higher for TunR2. Furthermore, TunR2 had a significantly longer half-life, as well as lower values for distribution rate constants ($K_{12}$ and $K_{21}$), elimination rate constant ($K_e$), and clearance rate ($Cl_T$) compared to Tun and TunR1. Additionally, TunR2 has the highest area under the curve (AUC) values (see Table 1).

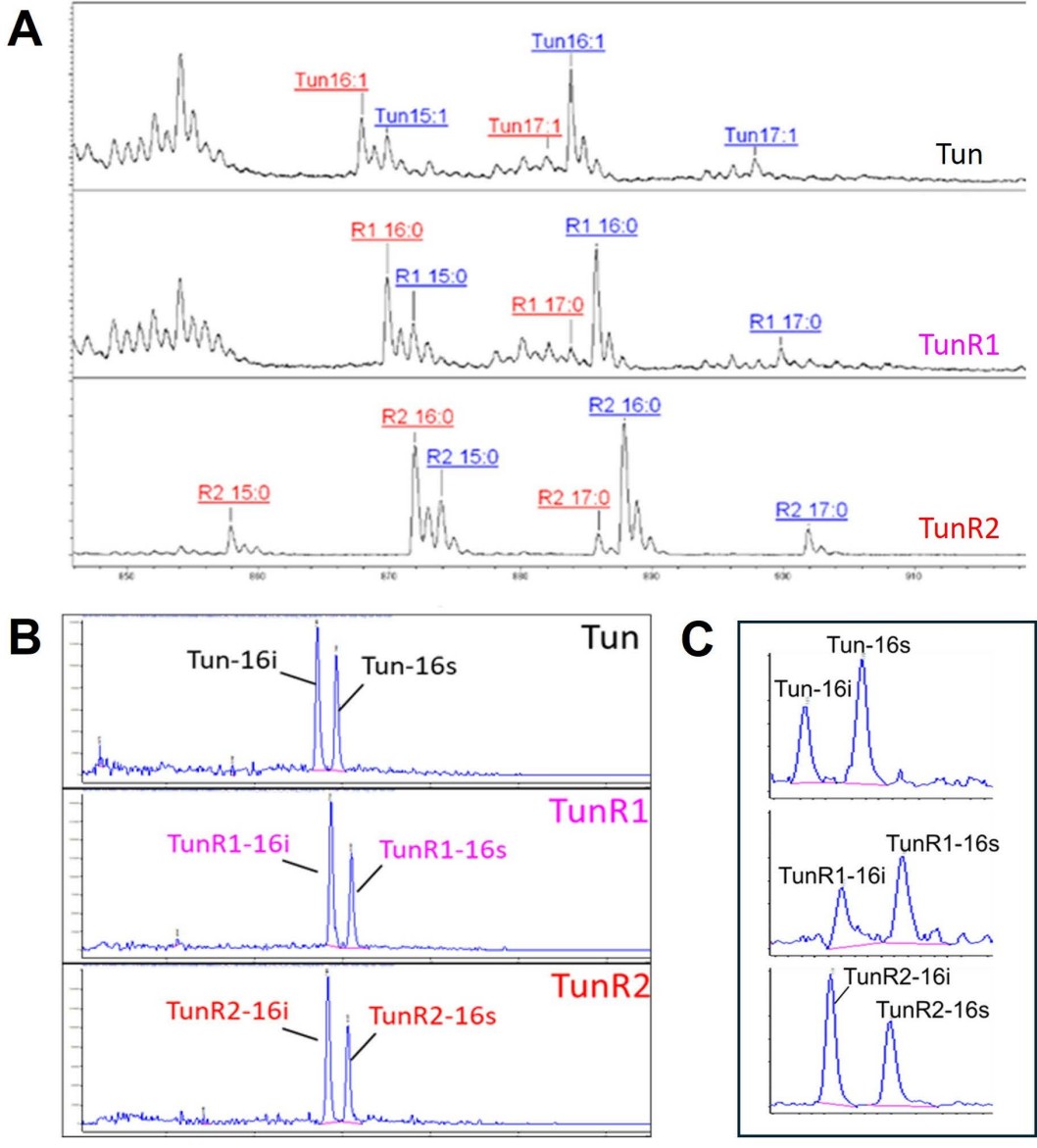

**Fig 4. Mouse pharmacokinetic study of Tun, TunR1, and TunR2 N-acyl variants in C57BL/6 mice (20 male & 20 female, 30uL of 10 mg/mL dose via tail vein injection).** Analysis of blood drawn 5 min. post-injection by MALDI-MS (A) and HPLC/MS (B). In the MALDI-MS graph, the labeled peaks refer to the various N-acylation forms of the drugs (15, 16, or 17 carbon atoms, with either 1 or 0 double bonds). The labeled ions in (A) are either [M+Na]+ (in red) or [M+K]+ (in blue). (C) LC/MS analysis of mouse liver extracts 24 h after inoculation with tunicamycin (top), TunR1 (middle), or TunR2 (bottom). The peaks are the 16 carbon Tun variants, with either *iso* (i) or straight chain (s) N-acyl substituents.

$Cp_0$, initial concentration that is equal to $C_{max}$, maximum concentration in plasma; V, volume (c: of the central compartment, t: tissue compartment, dss: distribution at steady state); K, rate constant ($K_{12}$: distribution from central to tissue/peripheral compartment; $K_{21}$: distribution from tissue to central compartment; Ke: elimination rate constant); $T_{1/2}$, terminal half-life (α: fast or distribution phase; β: slow or elimination phase; $K_{10}$: elimination); AUC, Area under the curve from time zero to 24h or infinite; MRT, mean residence time; Cl, clearance rate ($Cl_D$ distribution/intercompartment or $Cl_T$ total clearance). $R^2$: R squared value from the nonlinear fit bi-exponential curve (two-phase decay model).

On the other hand, when examining the variants individually, no significant differences were observed in $Vd_{ss}$ between C15, C16, and C17, but the C16 compound was the predominant, with significantly higher $Cp_0$ compared to C15 and C17. Moreover, the C16 is the variant with lower values of $Cl_T$ and MRT, and higher values of AUC, compared to C15 and C17 (S5, S6, and S7 Tables).

**Second mouse trial: Co-administration of oxacillin plus Tun, TunR1 or TunR2.** The second trial was a pilot study undertaken to evaluate the pharmacokinetics of a *beta*-lactam drug (oxacillin) in the presence of tunicamycin (either Tun, TunR1, or TunR2). This was done because *beta*-lactams are known to be greatly enhanced by tunicamycins *in vitro*, but this has yet to be demonstrated in a live animal. Forty mice (20M/20F) were given a single IV tail vein injection of either oxacillin (6.7 mg/mL, equivalent to 10.39 mg/Kg b.w. dose) or a corresponding amount of oxacillin plus either Tun, TunR1, or TunR2 (S3 Table). The oxacillin and the Tun components in these samples were evaluated by LC/ES-MS as described above. The oxacillin was cleared rapidly from blood samples, becoming undetectable after 15 minutes. Additionally, it was not found in tissue extracts after 24 h. As a result, the pharmacokinetic analysis of this drug was unable to be determined. However, the Tun components were detected in all of the blood samples, as noted in the first trial.

The same Tun, TunR1, and TunR2 *N*-acyl variants were also detected by LC/ES-MS in the organ samples collected after 24 hours, at which time the animals were culled. These compounds were most prevalent in the liver extracts (Fig 4C and 5B), and all the *N*-acyl variants of Tun, TunR1, and TunR2 in the liver were able to be quantified (Fig S5A-C Fig). There was a clear persistence of the compounds in the liver tissue, but in the case of heart and brain tissue, only Tun was detected, not TunR1 or TunR2 (Fig 5C-D and S5D-E Fig).

**Table 1. Comparative pharmacokinetics (PK) of Tun, TunR1, and TunR2 in C57BL/6 clonal mice.**

| PK parameters | Tun | TunR1 | TunR2 |
|---|---|---|---|
| $Cp_0 = C_{max}$ (µg/mL ± SD) | 54.4 ± 2.56 | 56.5 ± 3.42 | 60.0 ± 2.41[a] |
| V (L/kg ± SD) | **Vc:** 0.25 ± 0.012 <br> **Vt:** 0.57 ± 0.045 <br> **Vd$_{ss}$:** 0.82 ± 0.057 | **Vc:** 0.26 ± 0.016 <br> **Vt:** 0.34 ± 0.032[c] <br> **Vd$_{ss}$:** 0.60 ± 0.047[c] | **Vc:** 0.23 ± 0.092[b] <br> **Vt:** 0.25 ± 0.020[a, b] <br> **Vd$_{ss}$:** 0.48 ± 0.029[a, b] |
| K (h ± SD) | **$K_{12}$:** 2.53 ± 0.59 <br> **$K_{21}$:** 1.11 ± 0.29 <br> **$K_e$:** 0.77 ± 0.13 | **$K_{12}$:** 1.29 ± 0.34[c] <br> **$K_{21}$:** 0.97 ± 0.29 <br> **$K_e$:** 0.57 ± 0.17 | **$K_{12}$:** 0.56 ± 0.09[a] <br> **$K_{21}$:** 0.5 ± 0.1[a] <br> **$K_e$:** 0.42 ± 0.07[a] |
| $T_{1/2}$ (h ± SD) | **α** 0.16 ± 0.04 <br> **β** 3.39 ± 0.7 <br> **$K_{10}$** 0.89 ± 0.15 | **α** 0.26 ± 0.08 <br> **β** 3.28 ± 1.15 <br> **$K_{10}$** 1.22 ± 0.4 | **α** 0.52 ± 0.1[a, b] <br> **β** 4.3 ± 0.9 <br> **$K_{10}$** 1.65 ± 0.3[a] |
| $AUC_{0-24}$ (µg*h/mL ± SD) | 68 ± 4.07 | 90.05 ± 8.13[c] | 103.2 ± 3.92[a, b] |
| $AUC_{0-\infty}$ (µg*h/mL ± SD) | 70.07 ± 8.7 | 99.28 ± 26.32 | 143.6 ± 19.9[a, b] |
| MRT (h ± SD) | 4.22 ± 0.83 | 4.08 ± 1.05 | 5.03 ± 0.75 |
| $Cl_{rate}$ (mL/h/Kg ± SD) | **$Cl_D$:** 637.8 ± 120.2 <br> **$Cl_T$:** 195.5 ± 23.2 | **$Cl_D$:** 336.1 ± 69.7[c] <br> **$Cl_T$:** 147.9 ± 35.2 | **$Cl_D$:** 132.8 ± 17.3[a, b] <br> **$Cl_T$:** 99.6 ± 13.2[a] |
| $R^2$ | 0.9983 | 0.9926 | 0.9783 |

Accumulation in blood from the first mouse trial that received 30uL of 10 mg/mL via intravenous (IV) tail injection (in DMSO carrier). PK data, from two males and two females per time evaluated, was analyzed with a two-compartment model (2-phase decay equation, fast or distribution phase and slow or elimination phase). Statistical analysis with One-way ANOVA and Tukey's multiple comparisons test. Significant differences (denoted by "a") between the Tun and TunR2 groups ($p < 0.05$); Significant differences between TunR1 and TunR2 ($p < 0.05$) are indicated with "b"; Significant differences between Tun with TunR1 ($p < 0.01$) are indicated with "c".

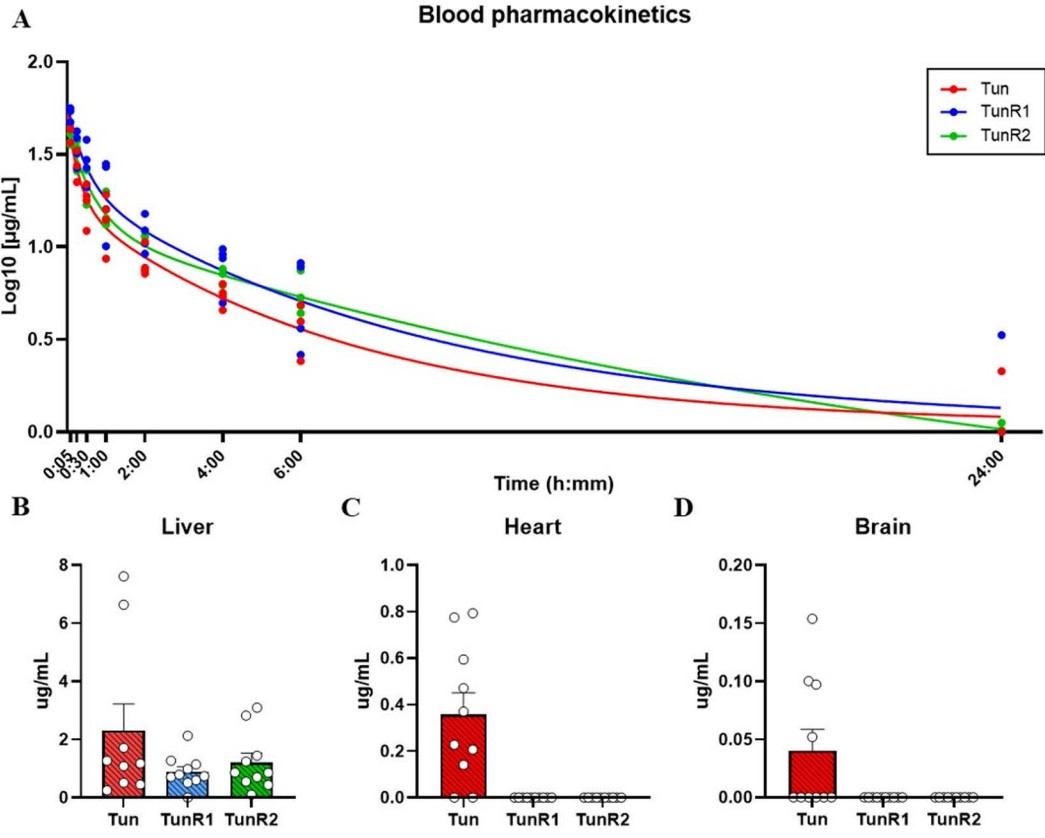

**Fig 5. Comparative pharmacokinetics of Tun, TunR1 and TunR2 in C57BL/6 clonal mice.** (2 males & 2 females per time evaluated) (A) Accumulation of each tunicamycin compound in mouse blood from the first trial that received 300µgdose via intravenous (IV) tail injection (in DMSO carrier). Concentration, expressed in Log10 *vs.* time, plots display observed points (filled circles). The solid line follows the two-phase decay nonlinear regression curve (two-compartment model fit). B-D) Detection of Tun, TunR1, and TunR2 in organ tissues (B-liver, C-heart, and D-brain) in mice from the second trial that received 201ug of Tun variant+Oxacillin via IV tail injection (in DMSO carrier). Drug concentration expressed in median (ug/mL) ± interquartile range. Each open circle symbol represents a replicate. The statistical analysis was performed by using Kruskal-Wallis Test and Dunn post-hoc test. No statistical differences in drug concentration in Liver. TunR1 and TunR2 were not detected in heart and brain tissue.

## Exploratory study of TunR2 in Holstein dairy cattle

**Pharmacokinetics.** Holstein dairy cows were housed in a BSL-2 barn facility, at the USDA-ARS National Animal Disease Center, for 90 days. Three animals (numbered 5327, 6739, and 6878) received three 5 mL intravenous bolus injections of 100 mg, 50 mg, and 45 mg of TunR2 on days 1, 3, and 5 of the study. A fourth animal (12802) received a control injection of the DOC alone. Cow 6739 was the only one lactating. Blood, urine, feces, and milk samples were collected and analyzed using MALDI/MS and LC-ES/MS (Fig 3B). Cow 6878 developed advanced clinical paratuberculosis and was culled on Day 17, but this was unrelated to the TunR2 treatment. The other cows, including controls, were culled after 90 days.

Four *N*-acylated forms of the TunR2 (TunR2-16-*iso*, TunR2-16-*straight*, Tun-R2-17-*anteiso*, and TunR2-17-*iso*) were detected in the blood samples 6h post injection (1st dose) as LC peaks at 7.5–9.5 min (S6A Fig). However, the initial concentrations were close to the limit of detection (LOD), so pharmacokinetic analysis to calculate the parameters could not be performed. The data were suitable for quantitative comparison of the different chain lengths and branching types (Fig 6A-B and S6A-B Fig). An example of the time-dependent accumulation and clearance in blood after the third bolus

(45 mg) of TunR2 is shown in S7 Fig. TunR2 exhibits similar behavior in blood and milk, with a peak concentration at 6h after each dose. Complete clearance of all variants from the bloodstream occurred within 10 days and between 10–20 days in the milk sample. No TunR2 compounds were detected in urine samples at any time during the study.

A similar chemical analysis was performed on the fecal samples, enabling TunR2 concentration measurements over the time course of the study (Fig 6C and S6C Fig). However, there was a higher accumulation of the TunR2 components in the fecal samples than in the blood or milk samples, with the initial appearance occurring 4–48 hours after each

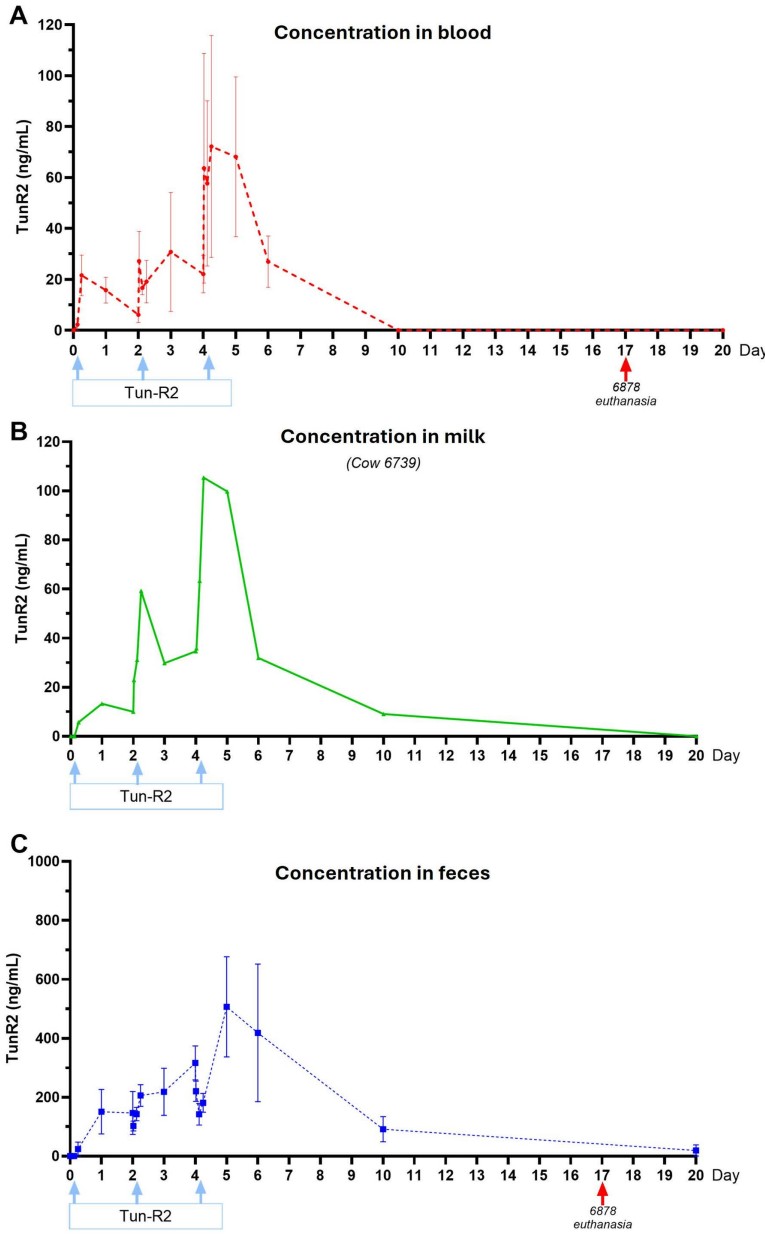

**Fig 6. TunR2 detection over time in three Holstein cows.** Drug concentration in blood (A), milk (B), and feces (C) samples at different time points after three bolus intravenous jugular TunR2 injections (100 mg, 50 mg, and 45 mg, marked with light-blue arrows). Milk was only obtained from cow #6739. The analysis was accomplished by LC/ES-MS. TunR2 was not observed in any of the urine samples.

injection (Fig 6C). This was cumulative, with the major amounts apparent after the third injection, although this was also the smallest dose injected (45 mg).

Finally, a complete analysis was also made of methanolic extracts from seventeen different tissue types collected from the animals at the end of the study. For TunR2-treated cows (5327 and 6739) and the control (12802), after 90 days, at which time the TunR2 is cleared from all organs. Cow 6878 was culled earlier (Day 17) and also showed no accumulation of TunR2 in any of the tissues, including the liver tissue.

## Monitoring health condition

To determine the safety of the using microdoses of TunR2 from the time of administration until the study endpoint, several parameters were measured.

**Physiological parameters and signs of disease.** During the study, weight changes were minimal, with a slight trend toward weight gain, except in cow 6878, which was in an advanced clinical stage of Johne's disease (JD) (S8 Fig). The animals did not show fever.

Mild inflammation at the injection site was the only post-dose side effect observed in TunR2-treated cows, not in control cow 12802. Cow 6878 had diarrhea due to JD's advanced disease status from day six and anorexia from day ten, leading to euthanasia on day seventeen for animal welfare reasons.

Cow 6739 developed mild submandibular edema and diarrhea from day sixty, while cows 5327 and 12802 showed no signs of JD.

**Complete Blood Count (CBC) test results.** The CBC results indicated that the control cow (#12802) had slightly elevated hematocrit (Ht) values with a normal red blood cell (RBC) count but showed no signs of dehydration. The TunR2 treatment did not affect RBC or Ht values, ruling out hemolytic anemia. However, cow 6878 had a higher reticulocyte count, suggesting increased erythropoiesis, likely due to systemic disease or increased erythropoietin from localized kidney hypoxia (S9 Fig).

Total white blood cell (WBC) counts were normal, though cow 6878 had elevated neutrophil counts on day −8. Cow 5327 displayed increased monocyte counts after TunR2 treatment, indicating a possible healing process from a chronic infection (S10 Fig).

The neutrophil-to-lymphocyte (N:L) ratio was elevated in cows 6739 and 12802 and highest in cow 6878 during days -15 to 3. Cow 6878 also had a slightly higher platelet count and the highest platelet-to-lymphocyte (P:L) ratio, with cow 6739 showing increased P:L ratios as well (S10 Fig). This suggests the presence of possible inflammatory processes in both cows 6878 and 6739.

**Blood chemistry test summary.** Albumin and globulins are mainly produced in the liver, except for gamma globulins and some beta-2 globulins, which are made by lymphocytes. Globulins, including fibrinogen and acute-phase proteins, can indirectly act as markers of liver function and inflammation. All animals showed slightly elevated total plasma protein levels, with some having increased albumin and globulin levels. Cow 6878 had a lower albumin-globulin ratio after day 2 (S11 Fig). Panhyperproteinemia without dehydration and slight hyperglobulinemia may suggest an inflammatory response, particularly in cow 6878, which had the lowest albumin levels. Glycemia, cholesterol, and calcemia were normal, but cow 6739 had slightly lower glycemia and the lowest phosphatemia levels on specific days (S11 Fig).

To monitor hepatocellular injury or inducible hepatocellular changes, several enzymes and total bilirubin were measured. Post-TunR2 treatment, aspartate aminotransferase (AST), alkaline phosphatase (ALP), and gamma-glutamyl transferase (GGT) levels increased but remained within normal ranges, except for AST in cow 6878 on day 10. Total bilirubin rose in cows 6739 and 5327 on days 5 and 6 after treatment. Alanine aminotransferase (ALT) showed no significant increases and is not a reliable indicator of liver disease in large animals (S12 Fig).

Creatine kinase (CK), a marker for muscle injury, was normal, with cow 6878 showing the highest values. No significant changes were noted in pancreatic enzymes (lipase and amylase) as shown in S12 Fig.

Blood urea nitrogen (BUN) and creatinine levels were within normal range. Treated cows displayed a slight decrease in BUN after TunR2 treatment, normalizing by day 20—except for cow 6878, which was euthanized on day 17. Overall, the results suggest that TunR2 had minimal to no toxic effects on liver and kidney function (S13 Fig).

**Urinalysis findings.** Cows 6739 and 5327 showed low urine specific gravity (USG) values, indicating isosthenuria, until days 10 and 30, respectively. Cow 6878 also exhibited low USG after day 6, along with mild proteinuria from days 6–17, which coincided with a mild urinary tract infection (UTI) (S13 Fig).

Microscopic examination of urine from cows 6878 and 12802 revealed red blood cells (RBC), white blood cells (WBC), and bacteria, indicating urinary infections between days 0–3 for cow 6878 and days 1–10 for cow 12802 (S13 Fig). Cow 6739 had an abnormal WBC count on day 60, suggesting a possible urinary infection. No renal tubular cells or casts were observed in the urine of any of the animals.

**Necropsy and histopathology findings.** Each cow demonstrated macroscopic and microscopic signs of intestinal disease consistent with paratuberculosis. Enteritis was present in various segments of the small intestine, including the jejunum and ileum (S14 and S15 Figs). The inflammatory cell infiltrate varied, with lymphocytes, plasma cells, and eosinophils predominating in the jejunum of most cows. Cow #6878 also showed granulomatous inflammation characterized by epithelioid macrophages containing acid-fast bacilli (Ziehl-Neelsen stain) (S15A and S15B Figs). All other cows, except cow #12802, exhibited granulomatous and eosinophilic inflammation in the ileum, with intestinal lymph nodes showing epithelioid macrophages and multinucleated giant cells indicative of granulomatous lymphadenitis (S15 Fig).

All cows exhibited diffuse vacuolar degeneration in liver tissue, characterized by swollen hepatocytes compressing the sinusoids and both microvesicular and larger vacuoles (25–30 μm) present. Cow #6878 showed the most severe hepatic vacuolar degeneration and had periportal hepatic myofibroblast hyperplasia and fibrosis. Periportal lymphoplasmacytic inflammation was observed in varying degrees in all cows. Notably, hepatocyte vacuolar degeneration and pericholangitis were also present in untreated animals with Johne's disease (JD), as seen in cow #5466 (S16 Fig).

In terms of kidney findings, animals 5327 and 6739 exhibited mild membranous glomerulonephritis, while animals 6878, 12802, and 5327 displayed mild to moderate lymphoplasmacytic interstitial nephritis (S17 Fig).

In cows #6878 and #6738, splenic macrophages contained hemosiderin pigment, indicating hemosiderosis from increased erythrophagocytosis. Necropsy findings for cow #6878 revealed emphysema and areas of consolidation in the anterior-dorsal lung, along with signs of heart dilation. Cow #6739 had focal areas of consolidated lung, petechiae, and small nodules on the lung surface.

**Johne's disease status.** The shedding of *Mycobacterium avium* subsp. *paratuberculosis* (Map) in feces was assessed using fecal culture and IS900 fecal qPCR. Results indicated that the Map burden remained constant after treatment with TunR2 (see S8 Table and S18 Fig). Two animals, 6739 and 6867, exhibited increased bacterial shedding, with a statistically significant rise in cow 6739. Cow 5327 tested positive only on day 1. The control animal (12802) showed intermittent shedding. Cow 6878 had increased shedding by the study's end, but her milk samples were intermittently positive and negative for Map on days 60 and 91 (S18 Fig).

At necropsy, bacterial cultures and PCR were performed on tissues to detect Map presence in animals. Cow 6878 was the most affected, followed by cow 6739, with varying levels of Map detected. Animals 5327 and 12802 had low levels (S9 Table and S19 Fig). These findings align with the fecal results. Details on macro- and microscopic lesions associated with JD are provided in the next section.

## Discussion

Native Tun is a nucleoside antibiotic known to induce detrimental ER stress in animals and is commonly used as an inducer of acute kidney injury (AKI) [6,27]. In male C57BL/6 mice, low-dose injected Tun (0.5 mg/kg I.P.) primarily causes damage to the pars recta of the distal tubule of the kidney and induces ER stress and apoptosis in this region [6].

Prevention of Tun-induced ER stress with 4-phenylbutyrate (4-PBA) partially relieves renal injury, and C/EBP homologous protein (CHOP) knockout mice treated with Tun do not develop renal injury [6,27]. Native Tun toxicity has also been reported in cattle, sheep, and pigs with toxic oral dose > 0.5 mg/kg/day for more than 2 days [4]. Further veterinary pharmacology is known concerning corynetoxins (Cnx), a group of toxins related to native Tun but differing in the pattern and degree of *N*-acylation [28]. Cnx are the cause of annual ryegrass toxicity (ARGT) that could affect several animals and is lethal to pasture animals (horses, donkeys, cattle, and sheep), as well as pigs, rats, mice, guinea pigs, and chickens [28,29]. The oral lethal dose of Cnx in sheep is 3.2–5.6 mg/Kg and is cumulative from multiple smaller doses administered up to 2-month intervals [29].

The cellular mechanism of toxicity of Tun is well defined, with uptake occurring via the MFSD2A transporter leading to inhibition of the dolichol-phosphate *N*-acetylglucosamine 1-phosphate PNPT-family transferase (GPT, *DPAGT1*) in the endoplasmic reticulum (ER). This enzyme catalyzes the transfer of GlcNAc-1-P from UDP-GlcNAc to dolichol phosphate (Dol-P) to form Dol-PP-GlcNAc, and is the first step to form the *N*-glycans required for *N*-glycoprotein biosynthesis [30,31]. The resultant disruption of the *N*-glycosylation process allows unfolded glycoproteins to accumulate in the ER lumen, triggering a lethal unfolded protein response (*upr*) via activation of the ATF6f pathway and ER stress-induced autophagy [30,31].

Homologs of the eukaryotic PNPT-family transferases also occur in bacterial cells, with an enzymatic role in the biosynthesis of peptidoglycan and teichoic acids, essential components of the bacterial cell wall [32–34]. These include the MraY/WecA/TagO-family transferases, essential bacterial enzymes that are also targeted by native Tun, so that the resultant inhibition leads to damage of the cell wall and subsequent bacterial lysis [34,35]. They have also been validated as targets for antimycobacterial drugs [10,36,37]. Various structural studies have shown that the major contributor to Tun-MraY binding is the Tun *N*-uridyl group (Fig 1) and occurs via a facial π-π complexation to an essential phenylalanine residue in the active site of MraY [15,30,31]. For TunR2, the *N*-uridyl group is replaced by 5,6-dihydrouracil, which lacks the 5′,6′-double bond and is consequently non-planar and devoid of the pseudo-aromatic ring π-electrons (Fig 1). The loss of planarity of the TunR2 uridyl group results in attenuated π-π binding to the active-site Phe residue in the eukaryotic PNPT resulting in the lower toxicity of the TunR2 molecule [32]. The modified TunR2 also lacks the *trans*-2,3-double bond found in the *N*-acyl chain of native Tun, which results in a further attenuated interaction with eukaryotic PNPT proteins [14,15]. Importantly, these structural changes have a lesser impact on their antibacterial activity, and TunR2 therefore holds considerable promise as novel mode-of-action antibacterial agent [8–10].

The TunR2 analogs have significantly lower eukaryotic toxicity compared to native Tun in *Saccharomyces* yeast [9], eukaryotic cell lines (Chinese hamster ovary CHO and human MDA-MB-231) [9], a live insect model (fall armyworm, *Spodoptera frugiperda*), and embryonic zebrafish (*Danio rerio*) [10]. TunR2 has also shown efficacy in a mycobacteria/zebrafish disease model, with attenuation of the host unfolded protein response (*upr*) and lowering of the mycobacterial infection and the concomitant macrophage response [10]. In this study, only one mouse in the Tun group (10 mg/mL) died, while the others remained alive and showed no apparent signs of toxicity during the 24-hour observation period. This outcome is consistent with known effects of Tun, as toxic and lethal reactions typically begin to manifest within 2–5 days after administration [5]. The lethal dose for 50% of the test population ($LD_{50}$) in mice has been established at 2 mg/kg [38]. As a result, our study was unable to provide a comprehensive assessment of the drug's toxicity due to the limited duration of the experiment. However, one thing to note is that only natural Tun was detected in the heart and brain, which could be associated with greater toxicity due to the accumulation of the drug in these tissues, since it is known to be cardio- and neurotoxic [31,39–41]. Further studies are necessary to evaluate the toxicity of TunR1 and TunR2 in the mouse model. Although the cattle trial yielded some insights into the drug's toxicity, which will be discussed in more detail later, it was primarily an exploratory trial that used subtherapeutic doses. Consequently, it is crucial to conduct further research involving a larger group of healthy cattle receiving higher doses, as well as longer-term trials in mice to evaluate the safety of TunR2.

The current study presents an LC-SIM/MS method for quantifying Tun, TunR1, and TunR2 in biological samples. This method facilitates pharmacokinetic analysis and demonstrates the high protein binding (>99%) anticipated due to the drug's lipophilic characteristics. It is important to note that although albumin is the primary drug-binding protein in the serum, other proteins such as globulins, lipoproteins, and α1-acid glycoproteins are also capable of binding drugs. It is known that the percentage of drug-protein binding depends on the ionization and lipophilicity of the drug [28].

In the stock solution and biological samples from Tun, TunR1, and TunR2, the most abundant compound is C16-iso, followed by smaller quantities of C15-iso and C17-anteiso. Both C15-iso and C16-iso are particularly effective at inhibiting MraY [6], the enzyme responsible for bacterial cell wall synthesis, making their higher presence desirable. Notably, the C14 compound was not detected in any of the biological samples.

The pharmacokinetic measurements in mice showed that Tun and TunR1 had a high to moderate $Vd_{ss}$, and higher volume in the tissue compartment ($Vt > Vc$). This indicates that the drug tends to be distributed in tissue [42]. In contrast, TunR2 presented a significant lower $Vd_{ss}$ than Tun and TunR2 with similar distribution in the central and tissue compartments ($Vc \approx Vt$). This is also reflected in the rate constant of distribution between compartments ($K_{12}$ and $K_{21}$).

In relation to half-life ($t_{1/2}$) and clearance ($Cl_T$), TunR2 has a longer half-life and lower clearance than natural Tun, although there are no differences in MRT. Compared to oxacillin, which has an estimated $t_{1/2} = 7.8\,min$ (fast elimination) [43], the $t_{1/2}$ of Tun, TunR1, and TunR2 is high (53.4, 73.2, and 99 min, respectively). This moderate to high half-life is common in lipophilic drugs due to their low free fraction in plasma (resulting from high protein binding), high volume of distribution, and low clearance rate. [44].

In the study by Gabani et al. (2009), rats were intravenously administered 1 mg/kg of natural Tun [45]. They also observed a low clearance of C15, C16, and C17 (equivalent to homologs B, C, and D) and moderate volume of distribution. It should be noted that the carrier used for the drug is differs from than in this present study, and neither the two-compartment model nor statistical analysis was used for the PK analysis.

In a pilot study conducted on cattle, we demonstrated that TunR2 can be administered intravenously using deoxycholate (DOC) as the carrier. The drug was found in blood, milk, and feces, but not in urine, even at the earliest time points. This suggests that the drug is primarily excreted through the bile, where the drug is found in higher concentrations than in blood or milk samples. Additionally, the passage of the drug through the gastrointestinal tract did not appear to reduce its potency, as there was no evidence of degradation or metabolic modification during this process. Furthermore, the increase in drug concentration observed in blood, milk, and feces after the second and third doses, administered on days 2 and 4, indicates that the drug has a long half-life and low clearance. This suggests that some of the drug remains in the system and is not eliminated before the next dose is given. It could also imply that part of the drug is being reabsorbed through enterohepatic cycling or redistributed from tissues [46].

Regarding the safety of low doses of TunR2 in cattle, the treated cows showed no signs of toxicity. The only side effects during administration were temporary swelling at the injection site. No hemolysis or anaphylaxis was observed. It is known that the natural Tun has toxic cumulative effects in ruminants, producing neurological signs, liver and kidney toxicity, with an increase in the levels of aspartate aminotransferase (AST), alkaline phosphatase (ALP), total bilirubin, and creatinine [4,47,48]. In our study, renal and hepatic toxicity was undetectable, as determined by serum chemistry analysis and urinalysis, or extremely low, even after the third dose, which suggests no cumulative effect of TunR2 when used in microdoses. CBC and blood biochemistry results are within the expected values for adult cows with JD.

The findings from necropsy and histopathological studies in cows are inconclusive, as healthy animals were not included in the research. The lesions observed in the liver and kidneys were also present in untreated animals and cannot be attributed to TunR2. It should be evaluated whether increasing the dose and frequency of administration to reach the therapeutic dose could result in mild to severe side effects. The toxic and lethal dose of TunR2 remains to be determined, but the doses used in this study can be considered safe. Also, further studies are needed to evaluate the potential use of TunR2 as an antibiotic against bacterial diseases, and to investigate its synergistic effects in combination with β-lactams.

## Supporting information

**S1 Fig. Schematic diagram of two-compartment IV bolus model used for the analysis of PK parameters for Tun, TunR1, and TunR2 in the 1ˢᵗ mouse trial.** Cp, drug concentration in plasma (central compartment), Ct, drug concentration in tissue compartment; Vc, volume of the central compartment; Vt: volume of the tissue compartment; $Vd_{ss}$, volume of distribution at steady state; $K_{12}$, rate constant of distribution from central to tissue/peripheral compartment; $K_{21}$, rate constant of distribution from tissue to central compartment; Ke, elimination rate constant; $Cl_D$, rate of distribution/intercompartment; $Cl_T$ total clearance rate.
(TIF)

**S1 Table. First mouse trial.**
(DOCX)

**S2 Table. Ratio of the different compounds (carbon chain length) and *N*-acyl variants (*anteiso*, *iso*, and *straight*) in Tun, TunR1, and TunR2, expressed in percentage (%).**
(DOCX)

**S3 Table. Second mouse trial: co-administration of oxacillin with Tun, TunR1, or TunR2.**
(DOCX)

**S4 Table. General characteristics of cows enrolled in this study.**
(DOCX)

**S2 Fig. Relative sensitivity of TunR2/DOC detection by LC-SIM/MS in water and biological samples from cows (milk, blood, and feces).** 1ug/mL of TunR2/DOC was added to 1 mL of blood or milk and 1 g of feces to evaluate the sensitivity of detection of the drug, compared to water samples. Dry residues from blood, milk, and water samples were redissolved in 200 μL methanol and feces in 1mL methanol (5x less concentrated compared to the other samples). The TunR2 detection in blood by LC-SIM/MS is reduced in 5-fold (pink line). Drug could not be detected in biological samples with 0.5 ug/mL or less of TunR2.
(TIF)

**S3 Fig. TunR2 (1ug/mL) detection by LC-ES/MS in samples containing A) Bovine serum albumin (BSA) 40 mg/mL (pink line), fetal bovine serum (FBS) (red line) compared to sample in water (purple line).** B) TunR2 (1ug/mL) detection in BSA 20 mg/mL (purple line) and 40 mg/mL (pink line) compared to sample in water (red line). Note that LC-ESI-MS has the highest sensitivity for the detection of TunR2 in water, followed by samples containing BSA (20 and 40 mg/mL) and finally FBS.
(TIF)

**S4 Fig. Comparative pharmacokinetics of the different *N*-acyl variants of Tun (A), TunR1 (B), and TunR2 (C) in C57BL/6 mice (1st trial).** Graphs show the N-acyl variant concentration (μg/mL ± SEM) *vs*. time.
(TIF)

**S5 Table. Comparative pharmacokinetics of Tun homologs in mouse blood.**
(DOCX)

**S6 Table. Comparative pharmacokinetics of TunR1 homologs in mice blood.**
(DOCX)

**S7 Table. Comparative pharmacokinetics of TunR2 homologs in mice blood.**
(DOCX)

**S5 Fig.  A-C) Comparative concentration of the different N-acyl variants of Tun, TunR1, and TunR2 in the liver from C57BL/6 mice (2nd trial).** D-E) Comparative concentration of the different N-acyl variants of Tun in the heart and brain. TunR1 and TunR2 were not detected in those tissues.
(TIF)

**S6 Fig.  Comparative pharmacokinetics of the different *N*-acyl variants of TunR2 in blood (A), milk from cow 6739 (B), and feces (C).**
(TIF)

**S7 Fig.  LC-ES/MS analysis of blood samples obtained from Holstein cattle after treatment with TunR2.** Blood samples from Cow 5327 at different time points:- pre-injection of 3rd dose, 30 min, 3 h, and 6 h on day 4, 24 h (day 5), and 48 h (day 6), showing the chromatographic peaks of the N-acylated TunR2 variants: C16-iso (7.74 min retention time), C16-straight (8.26 min), C17-anteiso (8.78 min), and C17-iso (9.06 min). At the pre-injection of the 3rd dose time point (on Day 4) there is a small residual peak for TunR2-16-iso due to carryover from the prior bolus of the drug. After just 30 min, the four TunR2 components are well defined in the blood stream and reach a maximum at 3 h post-injection. After this time, the different N-acylated forms of the TunR2 drug are cleared at the same rate (that is by 6 h post-injection), and only residual amounts of the two major forms (TunR2-16-iso and TunR2-16-straight) are still biologically available in the bloodstream after 24–48 h. Similar kinetics were also observed for Cows 6739 and 6878, and no HPLC Tun components were apparent for the negative control animal (cow 12802 which received only DOC) as it shows in the graph inside the gray box shows the negative control (cow 12802) at 24 h on Day 5. LC peaks were monitored by SIM-MS of TunR2-16 (m/z 628) and TunR2-17 (m/z 642).
(TIF)

**S8 Fig.  *Physiological parameters on cows.*** (A) The weights of each animal in kilograms (kg) over the 91-day study. Note that the normal weight of an adult Holstein cow is approximately 680 kg (dotted line). The graph also displays the percent change in weight for each animal, with increases shown in green and decreases in red. (B) Rectal temperature (°C). Normal rectal temperature for a dairy cow is 38–39.3°C (black dotted lines). Values >39.5°C or <36.7°C are considered hyper and hypothermia (dashed lines red and blue).
(TIF)

**S9 Fig.  Complete blood count (CBC) test results.** Erythrogram and Thrombogram. (A) Red blood cell (RBC) counts. (B) Reticulocyte counting (immature RBC). Dashed red line marks the limit of values that are considered significantly high and indicates increased erythropoiesis in animal 6878. This animal is not suffering from anemia, so the cause of a high reticulocyte count may be due to systemic disease (cardiac, pulmonary) or even an increase in erythropoietin production due to localized hypoxia in the kidney, which may or may not be due to subclinical renal disease. (C) Hematocrit. An elevated hematocrit with a normal RBC count in the control animal indicates possible dehydration because it is accompanied by a slight increase in total plasma protein (see Figure 9). (D) Hemoglobin, with a pattern similar to the hematocrit. (E) Mean corpuscular or cell volume (MCV). This value represents the volume of the average RBC, that in this case all animals are normocytic. Mean corpuscular or cell hemoglobin (MCH) indicate the average amount of hemoglobin in the RBC, that in this case all animals are normochromic. (F) Mean corpuscular hemoglobin concentration (MCHC), this value depends on the hemoglobin concentration and hematocrit. (H) Red distribution width (RDW)-standard deviation (SD) and (I) RDW- coefficient of variation (CV) reflect the cell volume variation within the RBC, higher values indicate more variation. (J) Platelets counting. Cow 6878 has the highest values, close to normal limit. (K) Mean platelet volume (MPV) reflects the average volume or size of platelets but also could be increased by platelet clumps. At the bottom right are the legends for all figures. Normal values are between black dotted lines. For all panels, the TunR2 doses are indicated with light blue arrows and cow 6878 euthanasia prior to study endpoint is indicated with a red arrow. Absolute eosinophils and basophils count were normal in all animals
(TIF)

**S10 Fig. Complete blood count (CBC) test results: Leukogram.** (A) White blood cells (WBC) counting. (B) Absolute neutrophil count. (C) Absolute lymphocyte count. (D) Absolute monocyte count. Monocytosis without neutrophilia in animal 5327 could be caused by a healing process of a chronic infection. (E) Neutrophil to lymphocyte ratio (N:L ratio). Red dashed line shows the limit of a significant increase in neutrophils (ratio >2). (F) Platelets to lymphocyte ratio (P:L ratio). Both the elevated N:L and P:L ratio indicate a possible inflammatory response in animal 6878 [20]. (G) Absolute eosinophils count and (H) absolute basophils count were normal in all animals. At the bottom right is the legends for all figures. Normal values are between black dotted lines. For all panels, the TunR2 doses are indicated with light blue arrows and cow 6878 euthanasia prior to study endpoint is indicated with a red arrow. Absolute eosinophils and basophils count were normal in all animals.
(TIF)

**S11 Fig. Blood chemistry test results.** Related to energy metabolism: (A) Blood glucose concentration (glycemia) and (B) Cholesterol. Not significant decrease in glucose and cholesterol detected. Proteins in blood: (C) Total plasma protein, (D) Albumin, (E) Globulin, and (F) Albumin-globulin ratio. Panhyperproteinemia could be dehydration and a slightly increase in globulins due to an inflammatory response (hyperglobulinemia). Minerals: (G) Phosphate in blood (phosphatemia) and (H) Calcium in blood (calcemia). Lower levels of albumin (in cow 6878) or phosphate (cow 6739) could indicate loss by enteropathy. At the bottom right is the legends for all figures. Normal values are between black dotted lines. Red and blue dashed line shows the limit of a significant increase or decrease in the value of the analyte in blood. For all panels, the TunR2 doses are indicated with light blue arrows and cow 6878 euthanasia prior to study endpoint is indicated with a red arrow.
(TIF)

**S12 Fig. Blood chemistry test results (continuation).** *Related to liver function:* (A) Aspartate aminotransferase (AST). No significant increase detected (need to be above the red dashed line), except for cow 6878 at time pre-euthanasia (10 days). This enzyme could be increased due to muscle injury if it's accompanied with an increase in CK. (B) Alkaline phosphatase (ALP). Slight transitional increase at 24–48hs post last dose but remains within normal values. (C) Gamma-glutamyltransferase (GGT). (D) Total bilirubin. Slight transitional increase at 24–48hs post last dose. *Other:* (E) Alanine aminotransferase (ALT). Not significant increase detected (need to be above the orange dashed line). NOTE: ALT is not a useful indicator of liver disease in large animals due to low enzyme activity in liver tissue. *Related to cardiac and skeletal muscle:* (F) Creatine Kinase (CK). Also need to consider AST levels. *Related to exocrine pancreatic function:* (G) Lipase and (H) Amylase.At the bottom right is the legends for all figures. Normal values are between black dotted lines. Red dashed line shows the limit of a significant increase in the value of the analyte in blood. For all panels, the TunR2 doses are indicated with light blue arrows and cow 6878 euthanasia prior to study endpoint is indicated with a red arrow.
(TIF)

**S13 Fig. Urinalysis (A-F) and blood chemistry test results related to kidney function (G and H).** (A) Urine specific gravity (USG). Values between bottom dotted line and red dashed line is isosthenuria (not urine concentration or dilution), may indicate kidney damage if accompanied by proteinuria (see graph E-UPC) and the presence of tubular cells or cast in the urinary sediment. (B) Urinary pH. Only a significant decrease detected in cow 6878 at time pre-euthanasia (10 days). (C) Urine protein needs to be evaluated with (D) Urine creatinine using the (E) Urine protein to creatinine ratio (UPC). Slight transitional increase at 1–5 days post last dose, that could indicate mild and reversible damage, since the values then normalize, together with the USG. (F) Representative photo of Urinary sediment (40x) cow #12802 at day 5. The abundant presence of leukocytes (>100 cells per high power field/microscopy field at 400x (hpf)) pyocites and transitional epithelial cells can be observed. This animal had a urinary infection between day 1–10. Also, cow 6878 presented a milder urinary tract infection between day 0–3 and after day 10 (See Table S1). (G) Blood urea nitrogen (BUN) and (H)

Creatinine showed normal levels. The slight transitional decrease in BUN could be due to an increase in the urea excretion by urine.At the bottom right is the legends for all figures. Normal values are between black dotted lines. Red dashed line shows the limit of a significant increase in the value of the analyte in blood. For all panels, the TunR2 doses are indicated with light blue arrows and cow 6878 euthanasia prior to study endpoint is indicated with a red arrow.
(TIF)

**S14 Fig. Representative image of Ileum, cecal valve (ICV) and intestine.** Note the reddish tissue of #6878 indicative of inflammation.
(TIF)

**S15 Fig. Intestinal tissue sections of cattle with Johne's disease.** (A) Ileum (H&E, 10x) from TunR2 treated cow #6878 with diffuse intermediate-lymphocytic lesion with crypt abscesses. Bar = 200 μm (B) Jejunum (ZN, 60x) from TunR2 treated cow #6878 showing the presence of acid-fast bacteria in epithelioid macrophages. Bar = 50 μm. (C) Proximal ileum (H&E, 20x) from TunR2 treated cow #6739 with diffuse multibacillary lesion. Bar = 100 μm. (D) Jejunum (H&E, 40x) from control (DOC treated) #12802 with eosinophilic enteritis and less severe lesions of Johne's Disease focal. Bar = 50 μm.
(TIF)

**S16 Fig. H&E liver sections of cattle.** (A) Control cow without Johne's disease showing normal liver tissue (20x). Bar = 100 μm. (B) Cow 5466 with JD (no treatment with TunR2 or DOC) showing vacuolar degeneration of hepatocytes (20x). Bar = 100 μm. (C) TunR2 treated cow #6878 showing vacuolar degeneration of hepatocytes (40x). Bar = 50 μm. (D) Control (DOC treated) showing swollen enlarged hepatocytes (20x). Bar = 100 μm. (E) Morphology of portal tracts in TunR2 treated cow 6878 that shows moderate periportal fibrosis (20x). Bar = 100 μm. (F) TunR2 treated cow #6739 showing a lymphoplasmacytic pericholangitis/portal hepatitis (20x). Bar = 100 μm.
(TIF)

**S17 Fig. H&E kidney sections of cattle.** (A) TunR2 treated cow #5327 showing mild membranous glomerulonephritis with attenuated tubular epithelium (10x). Bar = 200 μm. Upper right corner: lymphoplasmacytic interstitial nephritis (20x). Bar = 100 μm. (B) TunR2 treated cow #6739 showing mild membranous glomerulonephritis (40x). Bar = 50 μm. (C) TunR2 treated cow #6878 showing mild lymphoplasmacytic interstitial nephritis (10x). Bar = 200 μm. (D) Control (DOC treated) cow #12802 showing moderate lymphoplasmacytic interstitial nephritis (10x). Bar = 200 μm. Upper right corner: mild neutrophilic tubular nephritis (20x). Bar = 100 μm.
(TIF)

**S8 Table. Map shedding in the feces by culture for 6–12 weeks on HEYM.**
(DOCX)

**S18 Fig. A- Map shedding in the feces by IS900 qPCR.** The graph shows the Ct values for IS900 qPCR (y-axis) on the different time point (in days, x-axis). The TunR2 doses are indicated with light blue arrows and cow 6878 euthanasia prior to study endpoint is indicated with a red arrow (on day 18). The graph shading key is shown at the lower right. Note that both milk (dashed line) and feces were tested for cow 6739. B- IS900 qPCR fecal Ct values for cow #6739. Data expressed as Ct (median ± interquartile range) from pre/during the treatment (days −15 to 5) and post treatment (days 6 to end point). Statistical analysis was performed using the non-parametric test Mann-Whitney (***p < 0.001). No significant differences were observed in the rest of the animals.
(TIF)

**S9 Table. Culture and PCR for the presence of Map in bovine tissues.**
(DOCX)

**S19 Fig. Representative image of feces and tissues culture in Herrold's Egg-Yolk Medium with Mycobactin J (HEYM).** A-Cow 6739-ovaries, B-Cow 6739 feces at day 60, C-Cow 6878 feces at day 0, and D-6878 uterus.
(TIF)

**S1 Appendix. PK equations to calculate drug concentrations and PK parameters.** Pharmacokinetic parameters were calculated by two-compartmental using a linear regression analysis and residuals method to calculate using Microsoft Excel 2024 using standard equations.
(DOCX)

**S1 Data Set. Mice trial raw data including all mice weight, Tun, TunR1, and TunR2 concentration in blood and tissue.**
(XLSX)

**S2 Data Set. Cattle trial raw data including dose and sampling hours, animal weight, temperature, blood and urine analysis, and TunR2 concentration in blood.**
(XLSX)

## Acknowledgments

The authors are grateful to Trina Hartman for expert technical assistance.

## Author contributions

**Conceptualization:** Michael A. Jackson, Neil P. J. Price, John P. Bannantine.

**Formal analysis:** María A. Colombatti Olivieri, Eric D. Cassmann, Neil P. J. Price, John P. Bannantine.

**Investigation:** María A. Colombatti Olivieri, Michael A. Jackson, Neil P. J. Price, John P. Bannantine.

**Methodology:** María A. Colombatti Olivieri, Michael A. Jackson, Neil P. J. Price, John P. Bannantine.

**Resources:** Michael A. Jackson, Neil P. J. Price, John P. Bannantine.

**Writing – original draft:** María A. Colombatti Olivieri, Neil P. J. Price, John P. Bannantine.

**Writing – review & editing:** María A. Colombatti Olivieri, Eric D. Cassmann, Michael A. Jackson, Neil P. J. Price, John P. Bannantine.

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
