## [Decision Letter · Decision Letter 0]

Dear Dr. Colombatti Olivieri,

Thank you for submitting your manuscript to PLOS ONE. After careful consideration, we feel that it has merit but does not fully meet PLOS ONE’s publication criteria as it currently stands. Therefore, we invite you to submit a revised version of the manuscript that addresses the points raised during the review process.

We look forward to receiving your revised manuscript.

Kind regards,

Yung-Fu Chang

Academic Editor

PLOS ONE

2. To comply with PLOS ONE submissions requirements, in your Methods section, please provide additional information regarding the experiments involving animals and ensure you have included details on (1) methods of sacrifice, and (2) efforts to alleviate suffering.

“The study was supported by the USDA Agricultural Research Service.”

5. We notice that your supplementary figures are uploaded with the file type 'Figure'. Please amend the file type to 'Supporting Information'. Please ensure that each Supporting Information file has a legend listed in the manuscript after the references list.

Reviewers' comments:

Reviewer's Responses to Questions

**Comments to the Author**

1. Is the manuscript technically sound, and do the data support the conclusions?

Reviewer #1: Yes

2. Has the statistical analysis been performed appropriately and rigorously?

Reviewer #1: Yes

3. Have the authors made all data underlying the findings in their manuscript fully available?

Reviewer #1: Yes

4. Is the manuscript presented in an intelligible fashion and written in standard English?

Reviewer #1: No

Reviewer #1: Overall the manuscript is clearly presented, written well and presents data that justify the study claims and conclusions. Statistical analysis of the data appear to pass rigor and the data is available throughout the paper and supplemental materials. The mass spectrometry data does not appear to be fully available outside of the figures in the publication, but this may not be relevant to this study. Public repositories for storing MS raw files do exist for depositing large quantities of data.

There are a few grammatical errors and typos that need to be corrected during revision. Some examples include at line 143, 203, and 346. There are probably other errors that should be proofread for, as well. An additional oversight is the fact that the abstract states that non-compartmental PK analysis was used but it was established in the methods section of the paper and shown in supplemental figures that a two compartment analysis was used to evaluate PK parameters. Another oversight appears to be Figure S3. Y-axis abundances are written as "Absorbance" but the figure legend indicates that the data is from ESI-MS which does not record absorbance.

While the data presented are supported and discussed in the text, some of the discussion of the mass spectrometry standard samples seems unnecessary and does not contribute to the overall scientific body of knowledge, specifically, discussing the need to analyze standards for Tun, TunR1 and TunR2 dissolved in biological matrix. This is a well known phenomenon in mass spectrometry and it is considered common practice to analyze standard curve points and QC samples that have been spiked into the biological matrix of interest. Unless this is a novel discovery specific to these Tunicamycin molecules that had not previously been published, this portion of the manuscript does not contribute to the overall impact of the paper. Additionally, the assertion that TunR2 passes into the feces structurally unmodified is an important claim. It is discussed in the text but no data is shown that backs up this assertion. Structural determination of various modifications is important and not easily done by mass spectrometry alone. If the authors wish to leave this assertion in the paper or discuss it, some data should be included that support this claim.

It would have been a nice addition to the paper to see a different beta-lactam drug besides oxacillin that could have been subjected to some sort of PK analysis. However, there could be many different reasons for the choice of oxacillin in the study and this is not a major concern.

The manuscript thoroughly studies and establishes promise for Tunicamycin analogs to be used as adjuvants for antimicrobials. The data examine the PK/PD of Tun, TunR1, and TunR2 in a mouse model and look for toxic buildup of the analytes in off-target tissues and examine the administration of the experimental molecules with another antibiotic. The PK/PD analysis is thorough and well done. Next, Tun, TunR1, and TunR2 are given to cattle in a well-controlled study at sub-therapeutic doses to study tissue accumulation and potential side effects of these molecules. The rationale given for cattle dosing and the analysis of the various routes of elimination and potential tissue accumulation in the animals is well reasoned and well done. Overall, this study established the merits of these molecules as non-toxic alternatives to normally toxic Tunicamycin and successfully make the case that further study is needed of these promising analogs. Furthermore, the study clearly examines and establishes that of the Tun analogs, TunR2 is likely the more promising candidate, which is a result that is sure to inform future studies.

**Do you want your identity to be public for this peer review?** For information about this choice, including consent withdrawal, please see our Privacy Policy

Reviewer #1: No

---

## [Author Response · Author response to Decision Letter 1]

20 Jun 2025

Dear Reviewer,

Thank you for the time and effort that you dedicated to providing constructive feed-back on our manuscript “TunR2, a novel mode-of-action tunicamycin-type antibiotic: pharmacokinetics in C57BL/6 mouse and Holstein cattle”. We incorporated the sugges-tions and hope that this version enhances our manuscript and could meet the PLOS one standards for publication. Please see below, the author answers to your comments point-by-point.

Reviewer #1 - Comments to the Author

Overall the manuscript is clearly presented, written well and presents data that justi-fy the study claims and conclusions. Statistical analysis of the data appear to pass rigor and the data is available throughout the paper and supplemental materials. The mass spectrometry data does not appear to be fully available outside of the figures in the publica-tion, but this may not be relevant to this study. Public repositories for storing MS raw files do exist for depositing large quantities of data.

There are a few grammatical errors and typos that need to be corrected during revi-sion. Some examples include at line 143, 203, and 346. There are probably other errors that should be proofread for, as well. An additional oversight is the fact that the abstract states that non-compartmental PK analysis was used but it was established in the methods sec-tion of the paper and shown in supplemental figures that a two-compartment analysis was used to evaluate PK parameters. Another oversight appears to be Figure S3. Y-axis abun-dances are written as "Absorbance" but the figure legend indicates that the data is from ESI-MS which does not record absorbance.

Author’s answer: we have reviewed and corrected typos and grammatical errors. We cor-rected the abstract and Figure S3.

While the data presented are supported and discussed in the text, some of the dis-cussion of the mass spectrometry standard samples seems unnecessary and does not contribute to the overall scientific body of knowledge, specifically, discussing the need to analyze standards for Tun, TunR1 and TunR2 dissolved in biological matrix. This is a well-known phenomenon in mass spectrometry and it is considered common practice to ana-lyze standard curve points and QC samples that have been spiked into the biological matrix of interest. Unless this is a novel discovery specific to these Tunicamycin molecules that had not previously been published, this portion of the manuscript does not contribute to the overall impact of the paper. Additionally, the assertion that TunR2 passes into the feces structurally unmodified is an important claim. It is discussed in the text but no data is shown that backs up this assertion. Structural determination of various modifications is important and not easily done by mass spectrometry alone. If the authors wish to leave this assertion in the paper or discuss it, some data should be included that support this claim.

Author’s answer: We agree with the reviewer that LC/MS is a fairly standard technique for the analysis of drugs, and we have therefore removed the specific details of the MS method-ology. However, we found that selective ion monitoring (SIM) of [M + H - 221]+ fragment ion peaks was needed to give sufficient sensitivity. The Tun compounds were not detectable by standard electrospray MS monitoring and hence the MS-SIM technique requires some ex-planation. MALDI-TOF/MS is less commonly used to assay small molecules, but we found it useful for rapidly screening multiple TUN components simultaneously without prior chro-matographic separation. We feel that some explanation for this is needed.

Also, we remove from the manuscript the statement that TunR2 passes unchanged since we only have LC-MS data as pointed by the reviewer.

It would have been a nice addition to the paper to see a different beta-lactam drug besides oxacillin that could have been subjected to some sort of PK analysis. However, there could be many different reasons for the choice of oxacillin in the study and this is not a major concern.

Author’s answer: We appreciate the reviewer’s insightful comment. Oxacillin was selected as the representative beta-lactam in combination with tunicamycin derivatives for in vivo PK evaluation because oxacillin had a very pronounced enhancement with TunR2 (32- to 64-fold) when we tested them together in vitro (Price et al, 2017). While we agree that the eval-uation of the potential synergism with other beta-lactams and their PK would be of great interest for broadening the scope of synergistic interaction, our initial goal was to establish a proof-of-concept with a single, well-characterized beta-lactam antibiotic.

Citation: Neil PJ Price, Trina M Hartman, Jiakun Li, Kiran K Velpula, Todd A Naumann, Maheedhara R Guda, Biao Yu and Kenneth M Bischoff (2017). Modified tunicamycins with reduced eukaryotic toxicity that enhance the an-tibacterial activity of β-lactams. Journal of Antibiotics 70, 1070–1077.

The manuscript thoroughly studies and establishes promise for Tunicamycin ana-logs to be used as adjuvants for antimicrobials. The data examine the PK/PD of Tun, TunR1, and TunR2 in a mouse model and look for toxic buildup of the analytes in off-target tissues and examine the administration of the experimental molecules with another antibiotic. The PK/PD analysis is thorough and well done. Next, Tun, TunR1, and TunR2 are given to cattle in a well-controlled study at sub-therapeutic doses to study tissue accumulation and po-tential side effects of these molecules. The rationale given for cattle dosing and the analysis of the various routes of elimination and potential tissue accumulation in the animals is well reasoned and well done. Overall, this study established the merits of these molecules as non-toxic alternatives to normally toxic Tunicamycin and successfully make the case that further study is needed of these promising analogs. Furthermore, the study clearly exam-ines and establishes that of the Tun analogs, TunR2 is likely the more promising candidate, which is a result that is sure to inform future studies.

Author’s answer: we sincerely thank the reviewer for their thorough and encouraging evalu-ation of our manuscript. We are grateful for the constructive feedback and support.

---

## [Editor Report · Decision Letter 1]

TunR2, a novel mode-of-action tunicamycin-type antibiotic: pharmacokinetics in C57BL/6 mouse and Holstein cattle

PONE-D-25-19128R1

Dear Dr. Olivieri,

We’re pleased to inform you that your manuscript has been judged scientifically suitable for publication and will be formally accepted for publication once it meets all outstanding technical requirements.

Kind regards,

Yung-Fu Chang

Academic Editor

PLOS ONE
---

## [Editor Report · Acceptance letter]

PONE-D-25-19128R1

PLOS ONE

Dear Dr. Colombatti Olivieri,

I'm pleased to inform you that your manuscript has been deemed suitable for publication in PLOS ONE. Congratulations! Your manuscript is now being handed over to our production team.

Kind regards,

on behalf of

Dr. Yung-Fu Chang

Academic Editor

PLOS ONE